



# Assimilation of SST data in the POSEIDON system using the SOSSTA statistical-dynamical observation operator

Gerasimos Korres[1], Dimitra Denaxa[1], Eric Jansen[2], Isabelle Mirouze[2], Sam Pimentel[3], Wang-Hung Tse[3] and Andrea Storto[2].

[1]Hellenic Centre for Marine Research (HCMR), Athens, Greece

[2]Centro Euro-Mediterraneo sui Cambiamenti Climatici (CMCC), Italy

[3]Department of Mathematical Sciences, Trinity Western University, Langley, BC, Canada

*Correspondence to*: G. Korres (gkorres@hcmr.gr)

**Abstract:** In spite of their long-standing availability, the optimal assimilation of sea surface temperature (SST) observations retrieved from infrared and microwave space-borne sensors is still challenging in oceanographic forecast systems. One prominent problem stems from the fact that ocean general circulation models do not resolve the diurnal variability of SST data as measured by satellites. In order to improve SST data assimilation schemes and enhance the exploitation of swath SST data, an observation operator capable of representing the SST diurnal cycle is introduced and called SOSSTA. Firstly, a one-dimensional turbulence model is used to produce a data set of upper ocean temperature profiles with corresponding skin and subskin SSTs. A canonical correlation analysis is then used to extract the maximally correlated modes of variability between temperatures at depth and skin/subskin SST, conditioned to atmospheric state (insolation and wind speed). These canonical correlations form the novel observation operator, which is implemented in the POSEIDON model forecasting system (Aegean Sea) to test the assimilation of daytime SST retrievals from the SEVIRI infrared radiometer. Comparison of misfits (off-line assessment) suggests that the new operator outperforms the mere use of the first model level to calculate SST innovations. Real-world data assimilation experiments indicate that the use of the SOSSTA operator is beneficial to the skill scores and in particular improves the sea surface height analysis and forecast skill scores, whose improvement is maintained throughout a one year long experiment.

## 1. Introduction

Sea surface temperature (SST) is a key-component of the air-sea interaction, greatly affecting the formulation of major physical processes in the global ocean from sub-daily up to decadal time scales. In particular, the diurnal variability of SST has triggered many recent studies that focused on the impact of the SST diurnal cycle in different areas. Such studies have revealed a significant effect of diurnal variability in weather or climatic phenomena (as summarized in Kawai and Wada, 2007) as well as in applications such as modelling the oceanic heat budget (Marullo et al., 2016). This implies the need to further



improve the SST description through Ocean General Circulation Models (OGCM), with potential advantages for prediction

systems at all temporal and spatial scales.

From an ocean observing system perspective, SST is one of the best observed parameters in terms of both spatial and

temporal sampling. Remotely sensed SST dates back to the launch of TIRos in 1978 and has continued through present with

the launch of several satellite missions, with both polar-orbiting and geostationary orbits. Complementary in-situ SST

observations from a variety of platforms such as buoys and drifters allow satellite retrieval calibration and bias correction

procedures (e.g. Emery et al., 2001).

Infrared sensors such as AVHRR or SEVIRI measure the skin temperature at about 10 μm of depth, whereas microwave

sensors such as AMSR2 measure the subskin temperature at about 1 mm depth. On the other hand, SST analyses such as

OSTIA (Donlon et al., 2012) or NOAA OIv2 (Reynolds et al., 2007) provide the foundation SST (nominally at 10 m depth,

i.e. at the shallowest depth where the temperature is not affected by the diurnal cycle). Low winds and high insolation may

lead to several degrees of diurnal warming (Gentemann et al., 2003; Merchant et al., 2008; Le Borgne et al. 2012).

The diurnal cycle is to some extent resolved by OGCMs (Marullo et al. 2014). However, the vertical resolution of OGCM

and the approximations in the model physics set limitations in fully capturing the diurnal variations. Usually, the shallowest

level in OGCMs, varying from tens of centimeters to several meters, represents temperature that may differ significantly from

satellite-derived skin or subskin SST during the day. Therefore, the majority of ocean analysis systems currently assimilate

night-time SST observations (e.g. Waters et al., 2015), which are assumed to approximate foundation SST (free of diurnal

variability, as defined by GHRSST in Minett and Kaiser-Weiss, 2012). Alternatively, skin SST prognostic schemes have been

implemented for use in diurnal SST data assimilation (While et al., 2017), although this procedure has never been inserted in

a fully 3D oceanic analysis scheme.

In recent studies (Pimentel et al., 2008; While and Martin, 2014; While et al., 2015), an attempt of developing dedicated

observation operator for satellite SST data has been done, by embedding surface flux corrections from a dynamical model of

these processes. The impact of using these improved observation operators was beneficial, but still requiring developments

and adaptations for real-world operational systems. Interest in SST observation operators and diurnal warming has been

reinvigorated recently because of the potentially crucial impact of SST observations in (coupled) Earth system data

assimilation. Being the observations at the air-sea interface, SST data represent the fundamental observing network to constrain

air-sea fluxes in coupled prediction systems. In this context, assimilation of SST data with a diurnal warming-aware operator

has been embedded in the NASA atmospheric data assimilation system (Akella et al., 2017; Gentemann and Akella, 2018).

Future developments in coupled data assimilation will likely foster improved schemes for optimally exploiting SST

observations.

The SOSSTA (Statistical-dynamical observation Operator for SST data Assimilation) project, funded by the Copernicus

Marine Environment Monitoring Service (CMEMS) has developed a statistical-dynamical observation operator for the

assimilation of daytime satellite SST observations. The observation operator serves the purpose of comparing observations

with model equivalent, in order to calculate the innovations needed by data assimilation systems. The construction of the



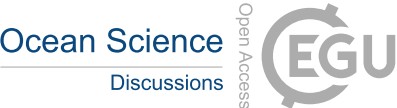
observation operator is based on the application of Canonical Correlation Analysis (CCA; Hotelling, 1936) to high resolution
diurnal cycle simulations from the 1D General Ocean Turbulence Model (GOTM), in order to statistically project sub-surface

profiles of temperature onto skin and subskin temperature. In particular, GOTM simulations produced (skin and subskin) SST
and upper ocean temperature data, which in turn were used in CCA resulting in a final set of maximally correlated empirical

relationships between temperature at various depths and skin/subskin SST equivalents, conditioned on atmospheric forcing.

     Here, data assimilation experiments performed with the observation operator developed within the SOSSTA project

(hereafter SOSSTA observation operator) are presented, ingesting skin SST data derived from the infrared sensor SEVIRI.
This paper focuses on the evaluation of the SOSSTA observation operator applied in the Aegean Sea modelling system (part

of the POSEIDON monitoring and forecasting system) and is organized as follows. Section 2 describes the formulation of the
SOSSTA observation operator. Section 3 provides an overview of the POSEIDON system and describes the implementation

of SOSSTA observation operator and the observational data used in the assimilation scheme. Section 4 presents and discusses
the results of the evaluation of the observation operator's performance in the Aegean in offline and online (data assimilation)

mode, while section 5 summarizes the conclusions of this evaluation.

## 2. The SOSSTA observation operator


     The SOSSTA observation operator, H, maps background temperature profiles, $x^b$, onto SST satellite observations (skin

and/or subskin, from infrared or microwave sensors, respectively), $y$, in order to compute innovations, $d$:

$$d = y - H(x^b). \tag{1}$$

The SOSSTA operator utilizes previously run GOTM 1D simulations, from which maximally correlated modes of
variability are extracted through canonical correlation analyses. The advantage of using a one-dimensional turbulence model

such as GOTM stems from its ability to be run at very high vertical resolution, thus sampling the near-surface ocean processes,
which would not otherwise be computationally feasible in OGCMs. In the following sub-sections, we outline the steps used to

construct the SOSSTA SST observation operator.

### 2.1 GOTM simulations

The General Ocean Turbulence Model (GOTM) is used to generate diurnal temperature profiles. These upper ocean
profiles form the training set on which the canonical correlations used by the observation operator are based. The ocean column

model resolves the vertical near-surface structure of the upper ocean; modelling temperatures to a depth of 75 m, but
encompassing 22 levels within the top 1m, including subskin SST and a skin SST. The model has been extensively tested and

validated over the Mediterranean Sea and found to match SEVIRI hourly SST with a RMSD of 0.62 °C and diurnal warming
magnitude with a RMSD of 0.66 °C [Pimentel et al., 2018]. For this study, several different training sets were produced based

on three different collections of GOTM simulations. (1) GOTM is run over a two-year period, 2013-2014, at 391 locations
(3/4-degree horizontal grid) covering the whole Mediterranean Sea. These simulations used MED-MFC daily mean reanalysis

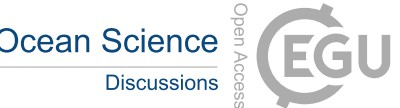

temperature and salinity profiles for daily initialization at local sunrise and were forced with 3-hourly ECMWF ERA-Interim

atmospheric reanalysis data. (2) A separate training set for the Aegean Sea region was also developed which is comprised of

GOTM simulations at 1/4-degree horizontal resolution, over the same two-year period. (3) GOTM is run over a 12-month

period 2016-10 to 2017-09 at 391 locations (3/4-degree horizontal grid) covering the whole Mediterranean Sea. For this set-

up the MED-MFC operational analyses are used to provide temperature and salinity profiles, from the hour closest to local

sunrise, for initializing the GOTM simulations and three-hourly ECMWF atmospheric forecast data is used for forcing. On

each occasion GOTM is initialized each day at local sunrise using the closest MED-MFC profiles and computes air-sea fluxes

and wind stress to model the full warming and cooling phase of the diurnal cycle, as well as the cool-skin effect of the viscous

skin layer. GOTM output is saved each hour.

## 2.2 CCA

Canonical Correlation Analysis (CCA, Hotelling 1936) is a technique to find cross-correlations between datasets. In the

case of SOSSTA the datasets consist of model temperature profiles on one hand, and SST (skin and subskin temperature) on

the other. CCA finds linear combinations of model temperature values and SST values in a way that each combination of

temperature is maximally correlated to the corresponding combination of SST. Moreover, each combination has to be

uncorrelated to all previous combinations. The set of linear combinations is what is referred to as the canonical variables. The

number of canonical variables is limited to the size of the smallest dataset, in this case two, as there are two values of SST

(skin and subskin temperature).

The relationship between the canonical variables is trivial to derive, as this is by definition a straight line with intercept at

zero. Therefore, canonical SST can be easily calculated from canonical temperature. Putting all of this together it is possible

to transform temperature into canonical temperature, calculate the corresponding canonical SST and use the reverse canonical

transformation to obtain SST itself. This method of parameterizing SST as a function of the model temperature has been shown

to lead to a good approximation of the GOTM results (Jansen et. al, 2018).

Formally, considering the transformation matrix $\mathbf{M} = \boldsymbol{a}\, D\, \boldsymbol{b}^{-1}$, where $D$ is a diagonal matrix of correlation coefficients

and $\boldsymbol{a}$ and $\boldsymbol{b}$ are the canonical coefficients such that $U = \boldsymbol{a}\, X'$ and $V = \boldsymbol{b}\, Y'$ are the maximally correlated canonical variables,

$X'$ and $Y'$ the mean-subtracted sets of temperature profiles and skin and subskin temperatures from GOTM, respectively. The

computed innovation using the SOSSTA observation operator is then given by:

$$\boldsymbol{d} = \boldsymbol{y} - \boldsymbol{x}^b\, \mathbf{M}(\boldsymbol{w}, \boldsymbol{r}) - \mathbf{K}(\boldsymbol{w}, \boldsymbol{r}),\qquad\qquad(2)$$

with $\mathbf{K} = \overline{Y} - \overline{X}\mathbf{M}$ an offset vector taking into account the mean values of $X$ and $Y$ denoted by the bar over the symbol. Here

we make explicit the dependence on the wind $\boldsymbol{w}$ and insolation $\boldsymbol{r}$ in the transformation.

Canonical correlations have been calculated separately, on an hourly basis, for 12 insolation and 8 wind categories, based

on the atmospheric forcing data available during the study period. This implies that the canonical correlations bear a

dependency on the atmospheric state at the sea surface, as well as the time of day. Details of atmospheric categories and

performances of the CCA are given in (Jansen et. al 2018).



### 3.  Implementation of SOSSTA observation operator in the POSEIDON analysis system

**3.1 The POSEIDON system**

The POSEIDON monitoring and forecasting system for the Greek Seas is established and periodically upgraded through
different research projects. One of the key ocean forecasting components of POSEIDON is the Aegean Sea modelling system.
It consists of a high-resolution implementation of the Princeton Ocean Model (POM) (Blumberg and Mellor, 1987) over the
Aegean Sea.  POM is a primitive equations ocean model, operating under the hydrostatic and Boussinesq approximations.
Model equations are written in sigma-coordinates and discretized using the centred second-order finite differences
approximation in a staggered "Arakawa C-grid" with a numerical scheme that conserves mass and energy. The model domain
covers the geographical area 19.5° E – 30° E and 30.4° N – 41° N with a horizontal resolution of 1/30°×1/30° and 24 sigma
layers along the vertical with a logarithmic distribution near the surface and the bottom.

The Aegean Sea model is forced with hourly surface fluxes of momentum, heat and water provided by the Poseidon - ETA
non-hydrostatic 1/20°×1/20° regional atmospheric model (Papadopoulos and Katsafados, 2009) issuing forecasts for 5 days
ahead. The net shortwave and the downward long-wave radiation terms are provided directly by the atmospheric model while
the upward long-wave radiation and the turbulent fluxes are calculated by the hydrodynamic model using its own SST and the
relevant atmospheric parameters (air temperature, relative humidity and wind velocity).

The model includes parameterization of fresh water discharge from major Greek rivers (Axios, Aliakmonas, Nestos,
Evros), while boundary conditions at the western and eastern open boundaries of the Aegean Sea hydrodynamic model are
provided on a daily basis (daily averaged fields) by the Copernicus CMEMS MED MFC system (Tonani et al 2014,
Dombrowsky et al. 2009).  Finally, the outflow of the Black Sea Water (BSW) in the Aegean Sea through the Dardanelles
Strait is described by an open boundary condition instead of the river parameterization used in the original version of the model
(Korres et al. 2003) and many other model implementations encompassing the Dardanelles Strait (see for example Kourafalou
and Barbopoulos, 2003; Androulidakis & Kourafalou, 2011; Oddo et al., 2014).

**3.2 The assimilation scheme**

The assimilation scheme used by the Aegean Sea forecasting system, is based on the Singular Evolutive Extended Kalman
(SEEK) filter which is an error subspace extended Kalman filter that operates with low-rank error covariance matrices as a
way to reduce the prohibitive computational burden of the extended Kalman filter (Pham et al., 1998). The filter is additionally
implemented with local approximation of the state error covariance for each updated state vector element and partial evolution
of the correction directions (localized SEEK filter – LSEEK). More specifically, a linear operator is defined that restricts the
global observation vector to its local part according to some cut-off radius (radius of influence) that is set (upon sensitivity
experiments) to 200 km in the way described in Korres et al. (2010). Innovations tapering is done using a 5[th] order polynomial
correlation function (Gaspari and Cohn, 1999). The LSEEK filter algorithm performs using a correction and a forecast step as
described in Korres et al. (2009).



In order to incorporate the SOSSTA observation operator, the original observation operator of the assimilation system had
to be extended. The new observation operator receives as input the daily mean atmospheric solar radiation, the daily mean
wind velocity and the time of the day (local time). From these data, the operator code selects the appropriate CCA and computes
skin (and subskin) SST by combining the CCA with actual values of the temperature model profile at a given observation
location. Thus, in order to implement the SOSSTA observation operator into the assimilation module of the model the
functionality of the original observation operator, which normally locates a set of eight model grid points surrounding a given
observation in space, has been extended. A set of four neighbouring profiles, ranging from 1.55 m to 44.5 m depth, are
identified and then interpolated to form a new temperature profile at the location of the skin SST observation. If this is
successfully done the procedure continues and finally a model equivalent skin SST is calculated in order to form the innovation
used by the SEEK filter. Skin SST observations located in model areas where the $1^{st}$ sigma layer is deeper than 1.55 m are
discarded as no valid temperature profile can be estimated by the interpolator. Considering the Aegean Sea model bathymetry,
the particular 24 sigma levels setup and the functioning of the interpolator of the SOSSTA observation operator, data
assimilation of skin SST observations can be performed only in the geographical areas shown in Fig. 12 no matter if skin SST
observations are available elsewhere in the model domain.

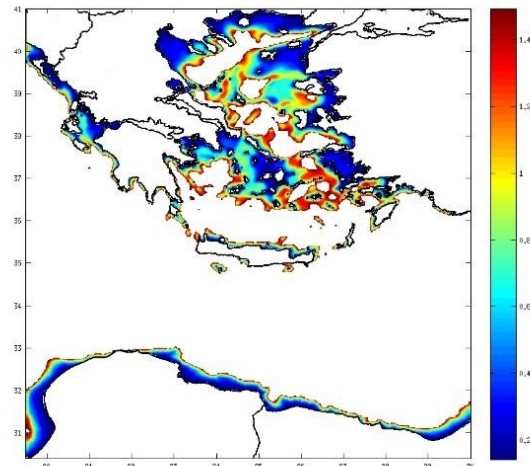

**Figure 1.** Depth of the first model sigma level, shown only in geographical areas (in color) where it is equal or shallower than
1.55 m




### 3.3 Observational datasets

In the standard operational cycle of the Aegean Sea forecasting system, a multivariate set of observations consisting of 1/8°
gridded AVISO SSH data, 1/16° gridded AVHRR foundation SST data (GOS Optimally Interpolated SST), T/S ARGO profiles
and temperature profiles from any available XBTs is assimilated. The observational data are assimilated on a weekly basis to
correct the forecast state of the Aegean model by projecting the innovation (misfit between the observations and the forecast
state in the observation space) into the state space using the time evolving filter statistics.

        The Sea Surface Height (SSH) observations from AVISO absolute dynamic topography are produced by Ssalto/Duacs
system and consist of gridded SSH data above geoid mapped on a regular 1/8° x 1/8° grid for the whole Mediterranean Sea.
SSH observations are assimilated in the POSEIDON system with a nominal accuracy of 3 cm.

AVHRR foundation SST data is the Copernicus foundation SST nominal operational product for the Mediterranean Sea in
NRT mode, in the form of daily gap-free maps (L4) at 1/16 x 1/16 horizontal resolution, processed by the ISAC-GOS system.
Data are obtained from infrared measurements collected by satellite radiometers and statistical interpolation. In order to avoid
diurnal warming contamination, only observations collected between 9 p.m. and 6 a.m. (local pixel time) are selected. When
assimilated into the model, the foundation SST data have a nominal accuracy of 0.8 °C and are considered to be at 6 m depth.
The latter means that foundation SST data are not assimilated in model areas where the depth of the first sigma level is below
6 m.

        A total of 811 quality checked temperature and salinity profiles were obtained from 20 Argo[1] floats fully or partially
operating in the geographical area of the Aegean Sea model during 2014. These observations are assimilated with a nominal
accuracy of 0.04 psu for the salinity profiles and a depth varying error for temperature profiles (0-6 m: 0.8°C; 6-20 m: 0.6°C;
20-100 m: 0.4 °C; 100-500 m: 0.2 °C; 500-bottom: 0.1 °C).

        In addition to these datasets, noon time (12:00 UTC) SEVIRI skin SST observations from MSG-3 (Meteosat Second
Generation) satellite are assimilated using the SOSSTA observation operator. Skin SST observations are calculated from the
infrared channels of SEVIRI every 15 minutes, with a spatial resolution of 4.5 km × 4.5 km on the Atlantic, the Mediterranean
and the Indian Ocean (geostationary satellite). For our experiments SEVIRI SST data have been retrieved for the period
01/01/2014 - 31/12/2014 for the whole Mediterranean Sea and at 3-hourly intervals for model assessment. The observational
error in the assimilation scheme was considered to be homogeneous and equal to 0.8 °C (further discussed in Sect. 4.2.1).

        Using only SEVIRI skin SST data characterized as "Excellent" or "Acceptable" in terms of cloud contamination
probability, mean summer and winter coverage was computed for the geographical areas where the first sigma level's depth is
equal or shallower than 1.55 m. Winter data availability was found to be significantly lower than summer giving a coverage
percentage relative to the theoretical maximum availability (considering observations at 3 hourly intervals) lower than 20 %.

---

[1] The Argo data were collected and made freely available by the International Argo Program and the national programs that contribute to
it. (http://www.argo.ucsd.edu, http://argo.jcommops.org). The Argo Program is part of the Global Ocean Observing System.



This spatially and temporally non-uniform distribution of useful SST observations was taken under consideration while analysing and interpreting results.

SST diurnal range computation in this work is based on the approach suggested by Sykes et al., 2011. According to this methodology, maximum and minimum SST values were considered valid when occurring between 11:00-21:00 and 01:00-11:00, local time respectively, where local time zones were defined as a function of longitude. Given that data at 3 hourly intervals were used for the model assessment, it was also considered necessary to discard any days with less than 5 valid recordings in order to capture the actual diurnal cycle.

## 4.  Evaluation of SOSSTA Observation Operator

In the scope of a first evaluation of different SOSSTA observation operator formulations, it was considered necessary to perform a series of offline experiments before proceeding with the SST data assimilation in the Aegean Sea model. Through this approach, strengths and weaknesses of different versions were identified, guiding within the SOSSTA project the development of new observation operators of improved performance. Several versions were constructed and tested, varying in the training dataset (different GOTM parametrizations), the observation operator's configuration for the dependence on the atmospheric conditions or the number of vertical levels required for its application.

Specifically, experiments showed that approximately 10 vertical levels ranging from 1.55 m to 44.5 m is the optimal choice for the model's temperature profile based on which the observation operator calculates the skin/subskin equivalents. Including greater depths did not improve the results which was to some degree expected, especially in summer months when stratification close to the sea surface is strong and deep waters sensitivity to surface variability is minimized. The observation operator sensitivity to atmospheric conditions was also examined leading to a final configuration of the atmospheric parameters in the operator, also concluding that using daily averages instead of instantaneous values of wind velocity and solar radiation does not decrease the operator's performance. Different Aegean model output were also tested in order to check if basic patterns of the performance of each operator's version are actually affected by slightly different model configurations. These experiments did not reveal important sensitivity to the different Aegean model experiments, thus the evaluation results discussed in this paper are based on one specific Aegean model configuration.

Offline and online (data assimilation) experiments presented in this paper are based on four selected versions of the SOSSTA observation operator hereinafter referred to as V1, V2, V3 and V4. All versions are based on the same GOTM configuration, which proved to sufficiently capture the vertical structure of temperature close to the sea surface as well as the skin SST diurnal variability in the Mediterranean.  However, the versions differ in regards to the forcing and initialization datasets used by GOTM. Version V1 is trained with the higher resolution 2013-2014 GOTM simulation limited in the Aegean and uses ERA-INTERIM data. V2 uses hourly ECMWF and Med-MFC data for 2016-2017, while V3 differs in that GOTM profiles are replaced with the temperature profiles taken from the Med-MFC model. GOTM in V3 is only used to model skin and subskin temperature, while the water column needed for the CCA computations is taken from the Med-MFC model.

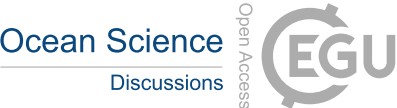

Finally, V4 is based on 2013-2014 GOTM data and ERA-INTERIM, same as V1 but using GOTM output for the whole
        Mediterranean and of the same spatial resolution as in V2 and V3.

## 4.1 Offline evaluation

### 4.1.1 Methodology

In the offline experiments 3-dimensional temperature data for 2014 from the 3-hourly Aegean model output and the
        corresponding atmospheric forcing fields were used. The procedure followed consists of:

(i)     Computation of mean daily values of solar radiation and wind speed in 2014 at each grid point of the Aegean model's
                domain

(ii)    Computation of new temperature profiles by interpolating the model's profiles at 10 specific depths ranging from 1.55 m
                to 44.5 m. In order to avoid errors induced by extrapolation only locations where the depth of the first sigma layer is less

252             than 1.55 m and the last greater than 44.5 m were used, thus several profiles needed to be discarded inevitably limiting
                the geographical area where the observation operator could be applied (see Fig. 12).

(iii)   Application of SOSSTA observation operator on each temperature profile, producing at each time-step a model equivalent
                skin SST value depending on wind velocity, solar radiation and local time.

(iv)    Application of a rejection criterion for the resulted skin SST values. Aiming to remove skin SST values of low confidence
                in an optimal way, a filtering tool was developed. For specific atmospheric conditions and local time, the highest and

258             lowest diurnal warming was computed for each profile used in the training dataset of the operator. These maximum and
                minimum allowable values, conditioned on local time, wind and insolation, were used to detect potential "unreasonable"

260             values. These values are rejected by the assimilation system under the consideration that the difference between the top
                model's temperature and the SOSSTA calculated skin SST for a particular wind-insolation-time combination should fall

262             within the range defined by these pre-calculated min/max values.

        Although the Aegean model does not produce skin SST, taking into account that the observation operator is applied only
at locations where the first sigma layer's depth is shallower than 1.55 m, inter-comparing this layer's temperature with skin
        SST satellite data is considered to be a valid approach. In this context, the observation operator's application is considered to
be beneficial as soon as SOSSTA skin SST produced through the offline experiments gives a lower RMS error with respect to
        SEVIRI than the model itself. Using only skin SST values of low cloud contamination probability led to a different number of
satellite observations corresponding to each time step to be included in the statistics.





### 4.1.2    Results and discussion


Over the year, the overall RMS errors for the first model level and the four versions of the observation operator with respect

to SEVIRI skin SST are presented in Table 1. We see that the offline application of every version is associated with a positive

impact. Results for all versions have been produced using the application of the rejection criterion which led to a removal of

approximately 3 % of the produced skin SST values.

Results from monthly analysis show that skin SST produced using the V2 SOSSTA observation operator presents lower

RMS error than the model in every month of 2014, thus revealing a satisfying offline performance of this version throughout

the year (Fig. 13). In this bar graph we see that in winter months the improvement resulting from the operator's application is

noticeably larger than during warmer months (July, August, September) when the operator's skin SST RMS error is still lower

but almost equal to the RMS error of the model.


**Table 1:** RMS error with respect to SEVIRI skin SST for V1, V2, V3 and V4 SOSSTA observation operators compared to

the first level's RMS error.

| SOSSTA Observation Operator | $1^{st}$ model level Temperature RMSE | SOSSTA skin SST RMSE |
|---|---|---|
| V1 | 0.9423 | 0.8766 |
| V2 | 0.9412 | 0.8380 |
| V3 | 0.9428 | 0.8408 |
| V4 | 0.9396 | 0.8206 |



This seasonal pattern can be partly explained as resulting from the seasonal variability of the RMS error of the initialization

data used by GOTM (Pimentel et al., 2018) as well as due to the Aegean model's better winter performance. The model is

expected to capture surface temperatures more sufficiently during winter months when the upper layer is well mixed and the

first sigma level's temperature (at depths less than 1.55m) is closer to skin SST values, while large skin SST diurnal variability

during warmer months is not expected to be properly described by a model level not representing the skin layer.





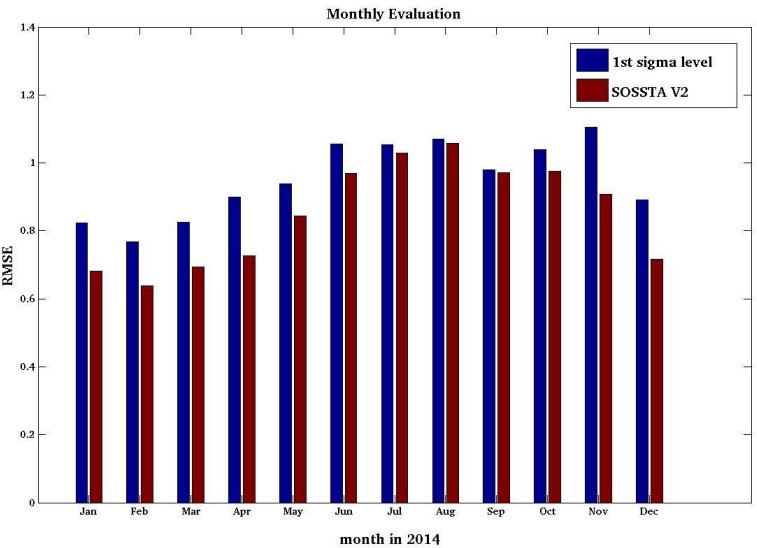


**Figure 2.** RMS error of the first level's temperature from the Aegean model (blue bars) compared to RMS error of the skin

SST computed using V2 (red bars), with respect to skin SST from all valid/available SEVIRI rerievals in 2014.
The seasonal behavior discussed above appears in all versions. However, V2 stands out as the only observation operator presenting lower RMS error than the model in every month of the year while there are months between July and October when
V1, V3 and V4 observation operators present slightly larger RMS errors than the model. As shown later in the data assimilation results (Sect. 4.2.2) it is indeed V2 that is found to bring the greatest improvement also in the online mode which is the actual
criterion for the final evaluation of the various versions.

    As for the RMS error spatial distribution, an improvement when the observation operator is applied can be seen in the

majority of the testing area. However, high RMS errors for the observation operator appear, mostly located in the north-eastern part of the Aegean close to the Dardanelles Strait, considered to be an area of high local complexities. Not applying the rejection
criterion here leads to a drop in improvement of between 1 and 3 percentage units but differences in RMS error temporal and spatial patterns were found to be negligible.
In order to focus on the observation operator's daytime offline performance SST values at 12:00 UTC were examined separately. Noontime results showed a slightly better offline performance of V2 than the other 3 versions. As with the overall
statistics for 2014, it is seen that in most areas the SOSSTA observation operator in offline mode achieves an improvement with respect to the model's performance, while there are areas in the north eastern Aegean as well as close to the western
African coasts of the domain where non negligible errors often appear.





### 4.2 Online evaluation

**4.2.1 Methodology**

A total of five experiments have been performed with the Aegean Sea modelling system over the period 1st Jan 2014 –
31st Dec 2014 in order to evaluate the performance of the four versions of SOSSTA observation operator and to estimate the impact of SEVIRI skin SST data assimilation into the Aegean Sea hydrodynamics. The experiments are listed and briefly
described in Table 2.
**Table 2:** Description of experiments (Control run – CTRL and experiments EXP1-EXP4) performed with the Aegean Sea modelling system over the period 1st Jan 2014 – 31st Dec 2014

| Experiment | Observation Operator | Observations | Frequency of data assimilation |
|---|---|---|---|
| CTRL | LSEEK | SSH, foundation SST (at 6m), Argo T/S profiles | Weekly |
| EXP1 | LSEEK including SOSSTA V1 | SSH, foundation SST (at 6m), Argo T/S profiles SEVIRI skin SST at 12:00 UTC | -Weekly for SSH, foundation SST and Argo T/S profiles -Daily for SEVIRI skin SST |
| EXP2 | LSEEK including SOSSTA V2 | SSH, foundation SST (at 6m), Argo T/S profiles SEVIRI skin SST at 12:00 UTC | -Weekly for SSH, foundation SST and Argo T/S profiles -Daily for SEVIRI skin SST |
| EXP3 | LSEEK including SOSSTA V3 | SSH, foundation SST (at 6m), Argo T/S profiles SEVIRI skin SST at 12:00 UTC | -Weekly for SSH, foundation SST and Argo T/S profiles -Daily for SEVIRI skin SST |
| EXP4 | LSEEK including SOSSTA V4 | SSH, foundation SST (at 6m), Argo T/S profiles SEVIRI skin SST at 12:00 UTC | -Weekly for SSH, foundation SST and Argo T/S profiles -Daily for SEVIRI skin SST |


In the control run of the model (CTRL) the Aegean Sea model is initialized at 1$^{st}$ January 2014 from the Aegean Sea
operational cycle and is integrated for a 12 month period (2014) nested with the MED MFC model (Tonani et al., 2014) while assimilating satellite SSH, foundation SST data at 6 m depth and Argo T/S profiles on a weekly basis using the localized SEEK
filter. In the control run of the model the SEVIRI skin SST data were retained from the assimilation process and are used later



as independent observations for the assessment of the daily system performance. Experiments EXP1 – EXP4, in addition to

the previous set of observational data, assimilate SEVIRI SST observations on a daily basis at 12:00 UTC using the four

different versions of the SOSSTA observation operator.

As already noted in Sect. 3.3, the observational error for SEVIRI skin SST is chosen to be homogeneous and equal to 0.8

°C. Such a selection of error although comparable to the SST diurnal range for the Aegean Sea (typical diurnal range varies

from 1°C in the North Aegean to 0.7 °C in the central part of the basin) is quite reasonable as it is the same as the foundation

SST error used in the assimilation system (there is no obvious reason to weight the usually sparse SEVIRI observations more

than the gridded foundation SST observations). It is also within the range of the SEVIRI observational error estimates (referred

to as Single Sensor Error Statistics) for the Aegean Sea which geographically varies between 0.7 – 1 °C as an annual mean for

year 2014.

In experiments EXP1- EXP4 the assimilation of SEVIRI skin SST observations takes place:

i) in areas of the model domain where minimum depth (depth of the first sigma level) is equal or lower than 1.55 m and

maximum depth higher than 41.28 m

ii) on days where no other observations (i.e. SSH, foundation SST at 6m and T/S profiles) are assimilated

iii) if the final number of available SEVIRI observations after applying criterion (i) is higher than 200.

It is important to mention here that the first of the above criteria excludes available SEVIRI observations in areas where

the first model sigma level is deeper than 1.55 m, since no appropriate temperature profiles can be found in these areas as input

for the SOSSTA observation operator. Such a requirement limits the number of observations available for data assimilation as

can be seen in Fig. 14. In this figure in black we show the initial number of SEVIRI observations at 12:00 UTC for each of the

207 days within 2014 where assimilation of SEVIRI observations is taking place, while the red curve corresponds to the final

number of available observations for data assimilation.  Only 8-14 % of the initial observations are used by the assimilation

system from May to September while for the rest of the year this number goes to 4-6 %. On the other hand, as the number of

observations is high (between $2.5 \times 10^4 – 3.5 \times 10^4$) during May to September, the system assimilates between 3500 – 5000

observations during this period though much less during the rest of the year.



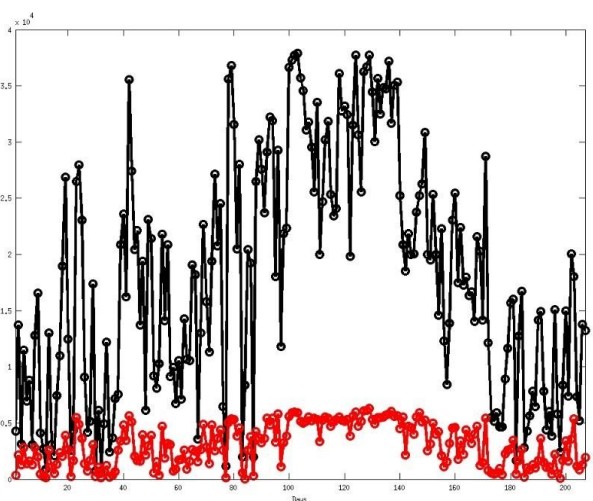


**Figure 3.** Number of available SEVIRI skin SST observations for each of 207 assimilation days within 2014. Black curve
corresponds to the original number of available observations within the Aegean Sea model domain while the red curve is the
number of observation after criterion (i) has been applied.

### 4.2.2 Results and Discussion

Forecast RMS error with respect to SEVIRI observations at 12:00 UTC for the control run and forecast and analysis RMS
errors for experiments EXP1 – EXP4 are shown in Table 3**Error! Reference source not found.** All experiments achieve RMS
errors lower than the control run with EXP2 (assimilation using the V2 SOSSTA Observation Operator) characterized by the
lowest values among the four.
**Table 3:** Overall forecast and analysis RMS errors with respect to SEVIRI skin SST observations at 12:00 UTC. Errors are
calculated in model areas where the depth of the first sigma level is lower or equal to 1.55 m


| Experiment | Forecast RMSE (ºC) | Analysis RMSE (ºC) |
|---|---|---|
| CTRL | 0.997 | - |
| EXP1 | 0.926 | 0.810 |
| EXP2 | 0.918 | 0.763 |
| EXP3 | 0.920 | 0.767 |
| EXP4 | 0.928 | 0.797 |





Considering that the V2 version of the observation operator has been constructed upon GOTM profiles initialized from hourly means of the operational Med-MFC run (instead of reanalysis daily means used in the V1, V3 and V4) and is forced
with the high resolution 2016-2017 ECMWF operational analysis (instead of coarser resolution ERA_INTERIM as in V1 and V4), it is expected to outperform the other versions of the observation operators, including V3 that uses the same forcing but
is characterized by the combination of Med-MFC system profiles with GOTM skin and subskin temperatures which may introduce incompatibilities due to the non-homogeneity of the temperature profiles data set used to build the operator.
As for the sensitivity of the results to the chosen observational error, an additional experiment was performed (not listed in Table 3), identical to EXP2 apart from the value of the SEVIRI skin SST observational error which was set to 0.5 °C. The
forecast RMS error for this experiment scored 0.916 °C (analysis RMSE 0.752 °C) while the rest of the statistical scores (SSH and overall temperature RMS errors) were slightly higher than those of the original EXP2. Based on these grounds and the
superior performance of the V2 operator, we decided to further analyse the EXP2 assimilation run behaviour and to show RMS errors with respect to other model variables.
Forecast and analysis RMS errors for skin SST, temperature (2 m – 1000 m) and SSH with respect to the corresponding observations (SEVIRI skin SST data at 12:00 UTC, satellite gridded SSH and foundation SST maps and Argo T/S) are shown
in figures 4-6 for CTRL & EXP2 experiments. Skin SST RMS errors are calculated on a daily basis (a total of 207 days) and for those model grid points where the depth of the first sigma level is equal or shallower than 1.55 m while the rest of the RMS
errors are estimated and presented every week (a total of 52 weeks). The forecast RMS error is computed just before the analysis update and can be considered an indicator of the consistency between the model dynamics and the filter statistics.



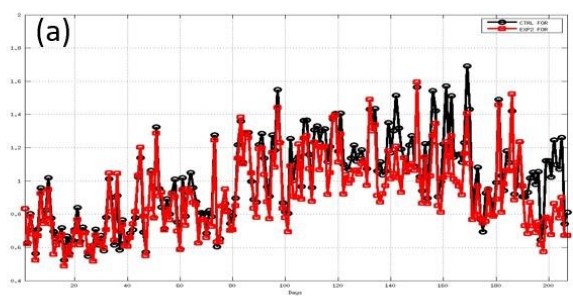

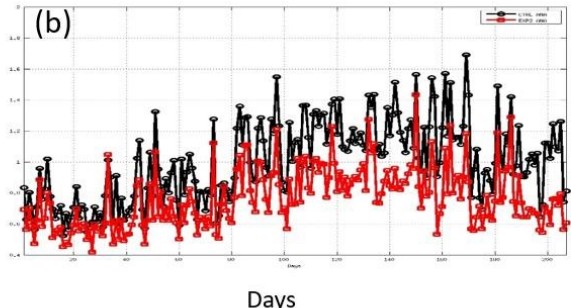

Days


**Figure 4.** Skin SST forecast (a) and analysis (b) RMS error corresponding to the model control run (black curve) and
experiment EXP2 (red curve) for the 207 days of SEVIRI skin SST data assimilation into the Aegean Sea model

For the first 100 days of SEVIRI data assimilation into the model (considering gaps this corresponds to the beginning of
May 2014), the skin SST forecast error is comparable with that of the control run (Fig. 15). Considering that skin SST has a
rather short memory as it is dynamically coupled with only a small fraction of the water column (mixed layer), such a behaviour
can be attributed to the limited number and the sparseness of the SEVIRI data that are available for data assimilation during
this period. For the remaining period, especially during summer when available SEVIRI observations for assimilation increase,
the forecast RMS error decreases with respect to the control run signifying a positive impact of data assimilation (Fig. 4). On
the other hand, the skin SST analysis RMS error is lower than the control run RMS error over the whole year with further
decrease during the period when the number of available SEVIRI observations increases. A similar behaviour is also observed
in the overall temperature (2 m – 1000 m) of the model domain (Fig. 16). After the 24th week (end of June 2014) of foundation





SST and Argo T/S profiles assimilation into the model, both forecast and analysis RMS errors become lower than those of the

control run showing clearly the impact of SEVIRI observations assimilation to the overall temperature behaviour of the model.


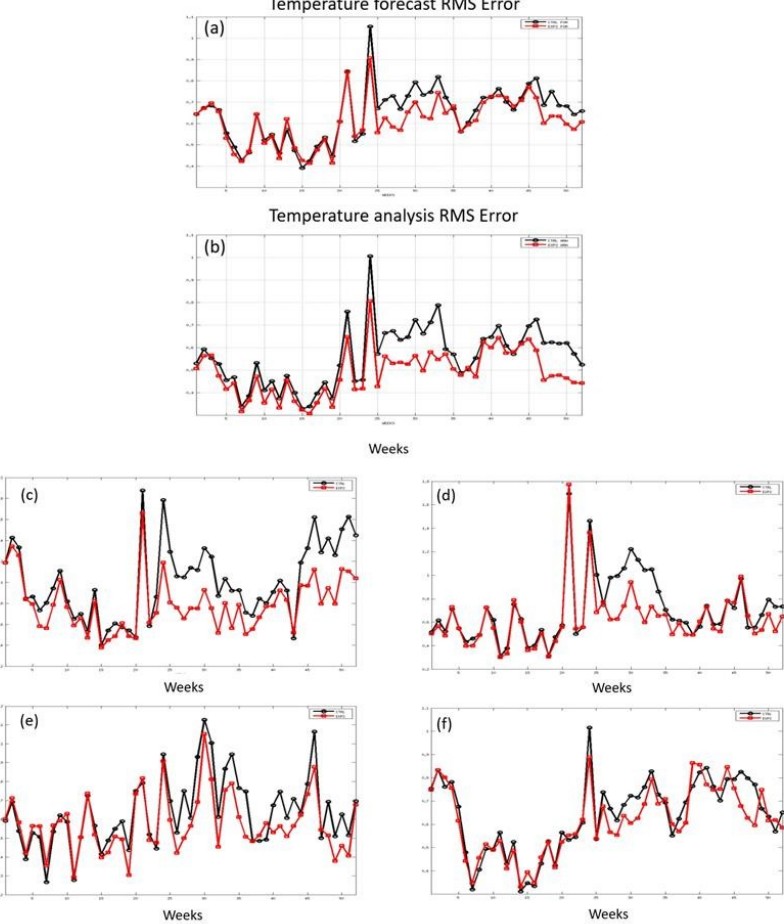

**Figure 5.** a) Temperature (2 m-1000 m) forecast RMS error corresponding to the model control run (black curve) and

experiment EXP2 (red curve) for the 52 weeks of SSH, foundation SST, Argo T/S profiles and SEVIRI SST assimilation into

the Aegean Sea model, b) same as in Fig. 16a but for the analysis RMS error, c) same as in Fig. 16a but for the north Aegean

(22° E - 27.5° E ; north of 39° N), d) same as in Fig. 16a but for the central Aegean (23 °E - 27.5° E ; 36.8° N - 39° N), (e)





same as in Fig. 16a but for the Cretan Sea (23° E - 27.5° E ; 35.5° N - 36.8° N), d) same as in Fig. 16a but for the north Libyan

Sea (21° E - 30° E ; 33.4° N - 35.4°N).


The spatial distribution of the annual mean (calculated over the 207 days in which skin SST assimilation took place in

EXP2) RMS error for skin SST is presented in Fig. 18. The error decrease for skin SST between the model control run and

EXP2 is very clear for the analyses fields while the forecasted fields show a limited error decrease. This is expected,

considering the behaviour of the skin SST RMS forecast error time series (Fig. 15). Thus the basin-wide mean RMS error of

the control run is 1.05 °C, which is reduced to 1°C for the forecasted fields of EXP2 and finally drops to 0.88 °C for the

analyses fields of EXP2. It is also noticeable that the forecast RMS error stays high over the north and north-eastern part of

the Aegean, the latter being related with the poor parameterization of Black Sea brackish waters inflow into the Aegean through

the Dardanelles Strait. Finally, the relatively high skin SST forecast and analysis RMS errors over the western and eastern

flanks of the northern African coasts are probably related with the functioning of the nesting mechanism of the Aegean Sea

model with the Med MFC model.

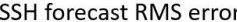

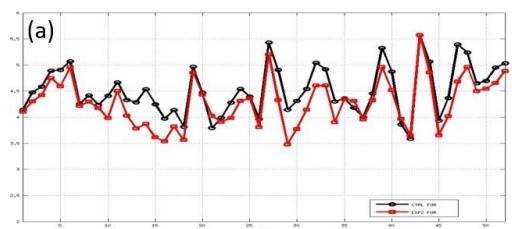

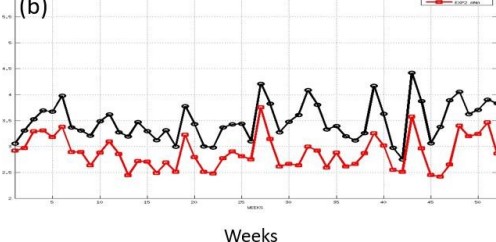

**Figure 6.** SSH forecast (a) and analysis (b) RMS error (in cm) corresponding to the model control run (black curve) and

experiment EXP2 (red curve) for the 52 weeks of SSH, foundation SST, Argo T/S profiles and SEVIRI SST assimilation into

the Aegean Sea model.





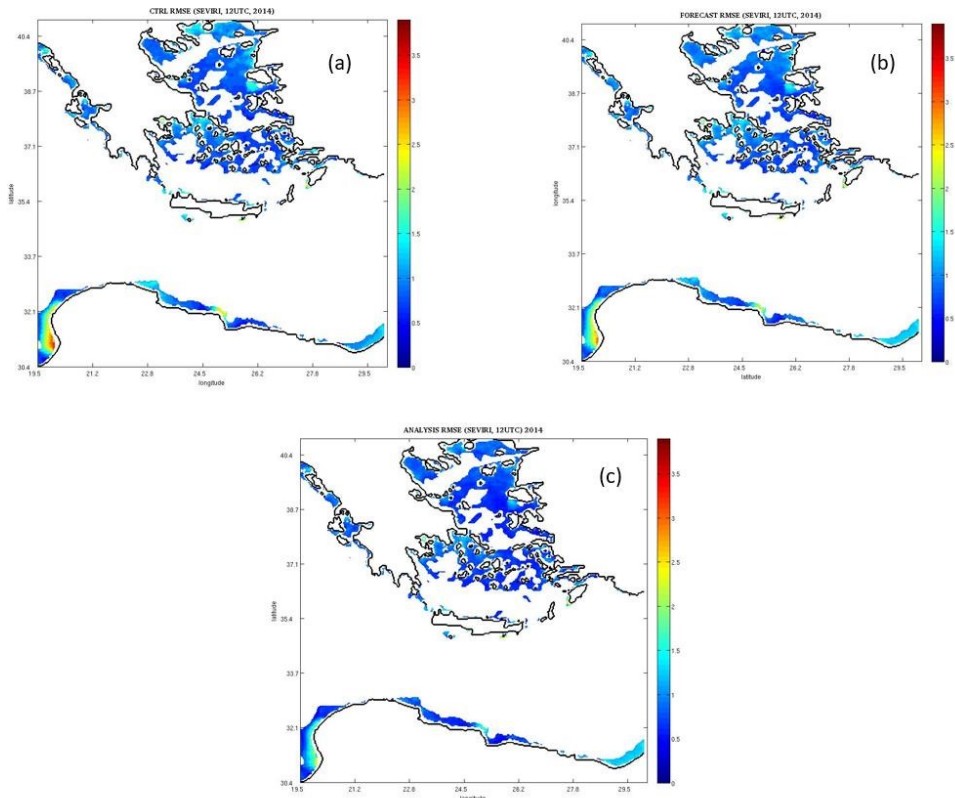

**Figure 7.** a) Skin SST RMS error corresponding to the model control run b) skin SST forecast RMS error corresponding to
EXP2 c) skin SST analysis RMS error corresponding to EXP2.


Another interesting index to monitor the efficiency of the Aegean Sea model and the effect of SEVIRI data assimilation in
EXP2 is the RMS error in estimating the skin SST diurnal range. The diurnal range error has been calculated (following the
methodology described in Sect. 3.3) for each of the 207 days when assimilation of SEVIRI observations is taking place (EXP2)
and is inter-compared to the one derived from the model control run. The mean error for the control run is 0.73 °C (diurnal
range of SEVIRI observations versus diurnal range of top model layer). This diurnal range RMS error is generally lower during
the winter period, when diurnal warming is less, and increases during summer and autumn period. The assimilation of skin



SST observations performed each day at 12:00 UTC is affecting the ability of the model to correctly simulate the diurnal range

of skin SST, decreasing the above error to 0.7 °C. Again the decrease of error is more effective from June 2014 onwards as we

have already seen for the skin SST and along the water column temperature of the model.

Regarding the SSH RMS error inter-comparison, a very interesting situation concerns the analysis error time series where

EXP2 is exhibiting an error decrease of approximately 0.5 cm with respect to the control run. This can be clearly seen in Fig.

19c and Fig. 19d where several error centres showing up in the control run (for example western and eastern off Crete island,

along the north African coast) drastically improve in EXP2 which shows error persistency only along the eastern and western

boundaries of the model domain where the open boundary conditions apply. The drastic improvement of analysis SSH error

is a direct consequence of SEVIRI skin SST assimilation (each time performed during the previous days and not coinciding

with SSH assimilation) and has consequences on other model variables (e.g. circulation pattern) also as it will be shown below.

It is interesting to discuss here that the analysis and forecast 2 m – 1000 m temperature RMS error behavior with respect

to the control run is highly compatible with the skin SST analysis error behavior (in both cases error starts decreasing by the

end of spring 2014 when more observations become available for data assimilation into the model). To better analyze the effect

of SEVIRI daily assimilation on the temperature field of the model, we present the overall temperature forecast RMS error

(EXP2 vs control run) for different areas of the Aegean Sea model namely the north Aegean, the central Aegean, the Cretan

Sea and the north Libyan Sea (Fig. 16). It is evident that the north and central Aegean Sea are impacted more by the assimilation

of SEVIRI skin SST as these two areas represent the places where the assimilation of SEVIRI observations takes place. The

Cretan Sea and the north part of the Libyan Sea represent model areas where the overall temperature field is impacted implicitly

by the SEVIRI assimilation taking place in the central and north Aegean. Implicitly here means that either the temperature

forecast RMS error improves due to the effect of SEVIRI assimilation to the global solution or by advection of water masses

with corrected characteristics.  For example, transport from the north and central areas to the southern parts of the Aegean Sea

as the dominant circulation pattern is cyclonic with water masses exiting the Dardanelles Strait moving initially to the south-

west and then southerly along the eastern coasts of Greece. Thus, although the assimilation of SEVIRI observations due to the

SOSSTA observation operator setup and the data availability involves only a limited area of the Aegean Sea (north and central

part of the basin) the circulation characteristics of the basin with the transport of the Black Sea water masses from the north-

east and to south-west flank of the basin can advect the positive impact of SEVIRI assimilation to non-assimilated (in terms

of SEVIRI observations) areas of the basin as for example the Cretan Sea.

On the other hand, the SSH analysis error of EXP2 is showing a totally different behavior compared to the temperature

field as it becomes lower than that of the control run even from the first two weeks of data assimilation of SEVIRI observations.

We believe that such a change between the two assimilation runs can be explained by considering the more frequent

observations fed into the data assimilation system performed in EXP2 (daily versus weekly in the control run) resulting in an

improved representation of the analysis error covariance matrix. The more frequent estimation of the Kalman filter analysis

error covariance matrix (in EXP2 this is done on a quasi-daily basis while in the control run on a weekly basis) by taking into





consideration all available and suitable SEVIRI observations for assimilation (in EXP2) proves to be beneficial at least for the SSH which with respect to other model state variables varies at shorter time scales.
When considering the maps of the SSH RMS error, the improvement in the analysis error covariance matrix has also global consequences as it is responsible for decreasing the SSH RMS error in geographical areas far from the areas where the SEVIRI
skin SST is taking place. We should once again emphasize here that skin SST data assimilation is done on a daily basis (upon data availability) but not coinciding with data assimilation of satellite SSH, foundation SST and T/S Argo profiles which is
performed strictly on a weekly basis.

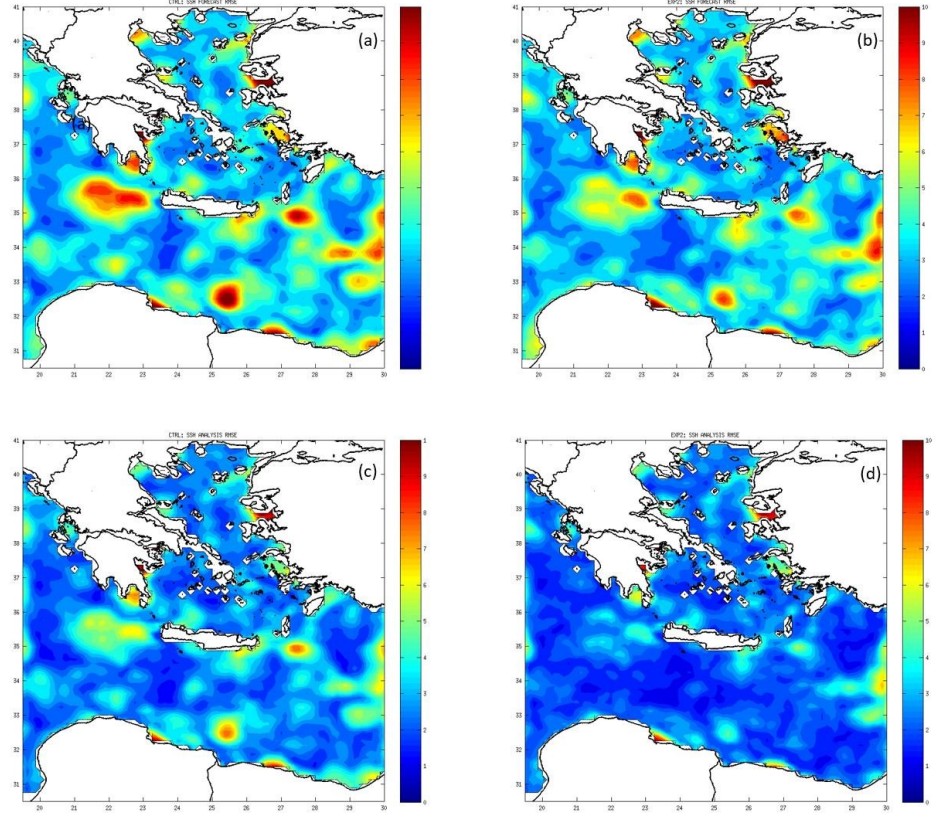


**Figure 8.** Forecast SSH RMS error (in cm) corresponding to a) model control run b) experiment EXP2. Analysis SSH RMS

error (in cm) corresponding to c) model control run and d) experiment EXP2.



The forecast SSH RMS error time series (Fig. 17) shows that for most of the time the introduction of SEVIRI data in EXP2
improves the results with respect to the model control run. However, considering the error reduction in the analysis fields this
improvement is not drastic as the overall error goes from 4.5 cm for the control run to 4.2 cm for EXP2. Again considering the
error maps presented in Fig. 19 we can observe a decrease of the RMS error in some areas; for example, the west and east
Cretan straits where the error of the control run is decreased by 2-3 cm.

To further show the impact of SEVIRI skin SST data assimilation into the Aegean Sea model state, we present, for 12
August 2014, the analysis SSH distribution corresponding to the model control run (Fig. 20a) and the model experiment EXP2
(Fig. 20b) together with the satellite image for the same date (Fig. 20c). The data assimilation performed in the model control
run does a very good job in bringing the solution close to reality, as seen by the satellite altimetry. However, some characteristic
details of this "reality" are missing from the control run solution, but show up in the analysis of EXP2, thus illustrating how
the assimilation of skin SST data in limited areas of the model domain is able to bring the Aegean Sea state closer to the real
world.  For example, the shape and strength of the Pelops anticyclone in the southwest of Peloponnesus, the structure of gyres
within the Cretan Sea (in particular of the large anticyclone in the Myrtoan Sea), the shape and strength of the Rhodes gyre
and the appearance of a dipole to the south of Crete.



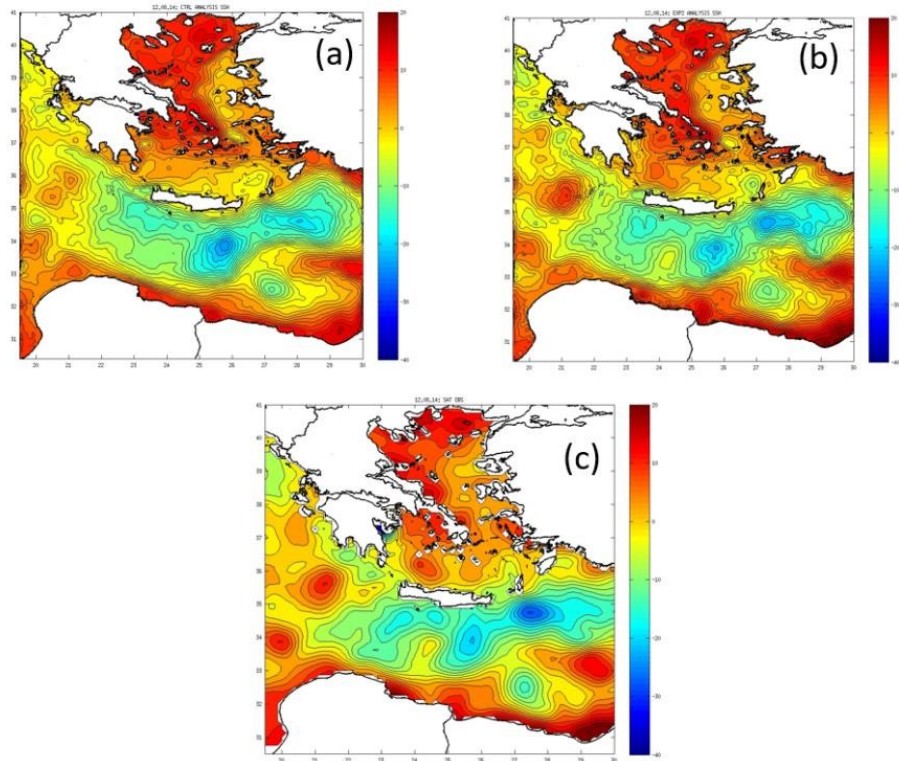


**Figure 9.** (a) Analysis SSH (in cm) for 12 August 2014 corresponding to CTRL run, (b) analysis SSH corresponding to EXP2,
and (c) satellite altimetry for 12 August 2014.
In addition to SSH, also other model variables are affected and benefit from skin SST data assimilation into the model. In

Fig. 21 we present the circulation pattern at 30 m depth corresponding to the analysis of the model control run and the analysis

of EXP2 for the 12[th] August 2014.  We also include a schematic of the Aegean Sea circulation (after Olson et al., 2007), in

order to have a reference of the general circulation and dynamical patterns that can occur in the Aegean Sea from the

climatological perspective, bearing in mind the interannual changes that can occur and modify this picture. A description of

the two circulation fields corresponding to the control run and EXP2, follows:

- A large anticyclone (also known as the Samothraki anticyclone) develops in the northeast of the Aegean Sea in both

model runs, although slightly shifted to the south-west with respect to the satellite altimetry.



- The intense boundary current that usually flows southwards along the Evia Island as part of the cyclonic general circulation that develops in the basin and brings Black Sea waters to the Cretan Sea through the Cyclades islands is totally
absent in both runs. However, in EXP2 we clearly see a well organised and intense southward flow, starting from the Dardanelles Strait, running in the middle of the Aegean and carrying the BSW into the Cyclades plateau and finally into the
Cretan basin.

   - In EXP2, the western part of the Cretan Sea and the Myrtoan Sea is occupied by a two-lobe and rather broad anticyclonic
circulation pattern consistent with the model sea surface height field and the satellite observations. This pattern is completely absent in the control run of the model.

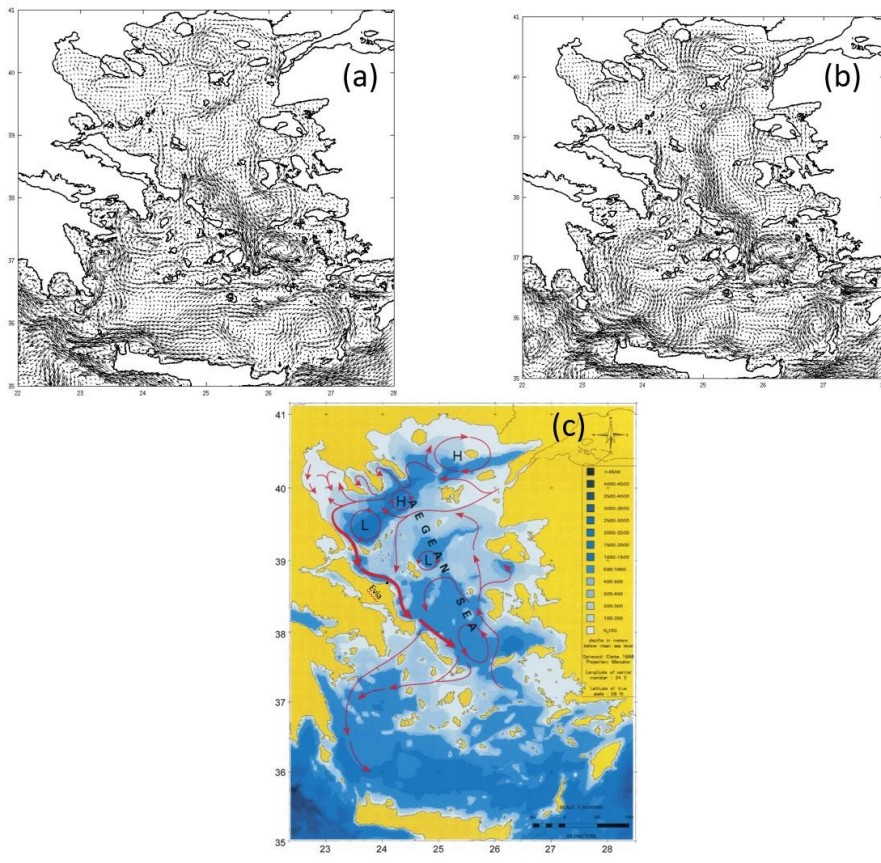


**Figure 10.** The circulation field at 30m depth on 12 August 2014 corresponding to (a) the model control run and (b) model
experiment EXP2. A schematic of the Aegean Sea general circulation after Olson et al., 2007, (c).



Additional changes introduced to the circulation field in the central Levantine basin outside the Aegean Sea (not shown here) involve the reshaping and intensification of the Pelops gyre of EXP2 with respect to the control run and modifications
to the Rhodes gyre system which during this period has a two-lobe structure encompassing a cyclone that develops south of the eastern part of Crete.
Another example of the effect of skin SST data assimilation to the Aegean Sea model state, concerns the spatial distribution of sea surface height on 8th April 2014. As with Fig. 20, we show the analysis SSH corresponding to the model control run
(Fig. 22a) and the model experiment EXP2 (Fig. 22b) together with the satellite altimetry (Fig. 22c) for the same day. The SSH analyses for both model runs are very similar to each other and capture the main dynamic structures depicted in the
satellite altimetry. However, careful inter-comparison shows differences between the two model analyses that can be attribute to the impact of daily data assimilation of SEVIRI skin SST performed in EXP2:
- The shape, orientation and strength of the Pelops anticyclone to southwest of Peloponnesus of EXP2 compares very well with the satellite altimetry. The same dynamic structure appears weak and shrunk in the control run of the model.
- The Rhodes gyre cyclone has a three-lobe structure in the satellite altimetry which is also correctly represented in shape, orientation and strength in the analysis of EXP2. On the other hand, although the control run of the model captures the same
cyclonic structure consisting of three lobes, the two northern lobes have a wrong orientation (western lobe south of Crete), shape (eastern lobe) and strength (both lobes) contrary to the observations and EXP2 analysis.
- The anticyclonic eddy at 25.5° E, neighboring the African coast, is almost missing from the model control run but it clearly shows up and has a well formed dynamic structure in the model experiment EXP2 matching that of the satellite altimetry
picture.

  - The signature of a large anticyclone in the west Cretan Sea is evident in EXP2 and satellite altimetry but is almost missing

from the model control run.




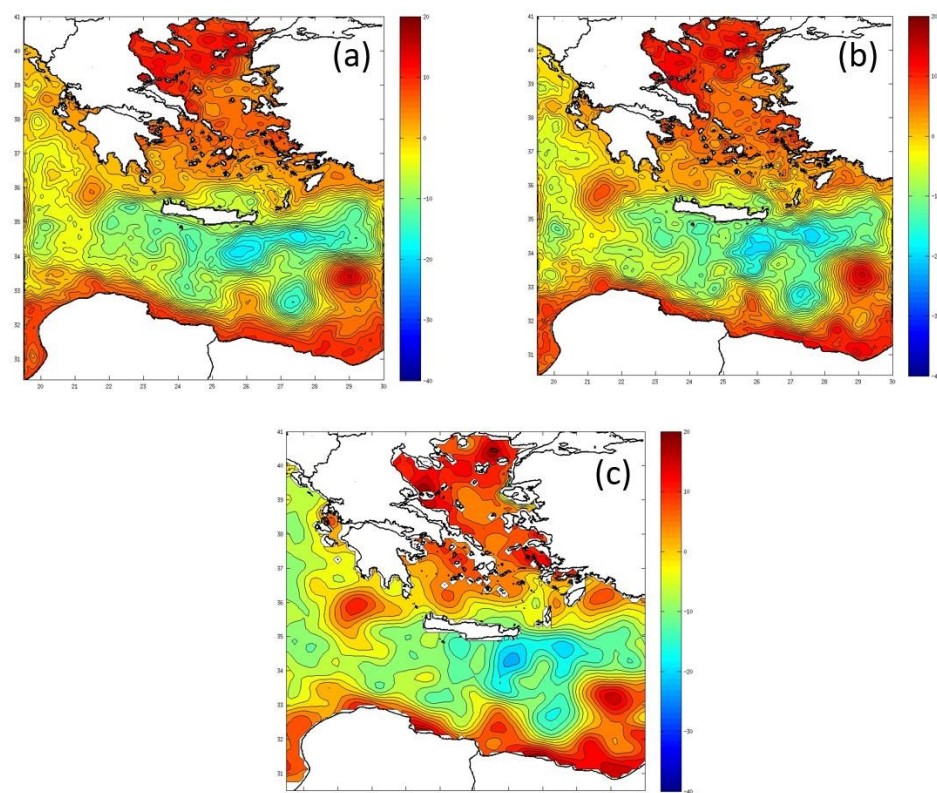


**Figure 11.** (a) Analysis SSH (in cm) for 8 April 2014 corresponding to CTRL run, (b) analysis SSH corresponding to EXP2,
and (c) satellite altimetry for 8 April 2014.
**5. Summary and Conclusions**
In this paper, an observation operator for satellite SST data assimilation able to account for SST diurnal variations and the
cool-skin effect is formulated. We first derive the statistical-dynamical observation operator, and later show the
implementation of this observation operator in the POSEIDON analysis system (Aegean Sea) and the evaluation of their
performance assimilating skin SST observations derived from the infrared sensor SEVIRI for 2014. The observation operator
is based upon simulations performed with the 1D column model GOTM, from which maximally correlated modes of variability
between skin/subskin SST and temperature profiles were extracted through canonical correlation analysis. Dependence on the



atmospheric state is introduced by forming as many canonical correlation sets as the combination of wind speed and insolation

categories designed for the Mediterranean Sea.

Given values for daily mean solar radiation, daily mean wind velocity and local time, the observation operator will project

a model temperature profile onto skin and/or subskin SST for data assimilation. The observation operator was initially applied

in an offline mode to the Aegean model profiles and produced skin SST equivalents for comparison to SEVIRI observations.

Considering that the application of the SST observation operator is beneficial as soon as the estimated skin SST versus

observations gives a lower RMS error than that from the first model level temperature, it was found that all versions of the

operator were able to bring an improvement compared to the model performance (without the operator) in the majority of the

testing area. However, non-negligible errors when using the operator can occur, mostly located in north-eastern Aegean off

the Dardanelles Strait which is an area of high complexity as well as close to the western African coast, most probably related

with the functioning of the nesting mechanism with the Med MFC model.

In the scope of the online evaluation, one control run and four assimilation experiments have been performed with the

Aegean Sea modelling system in order to demonstrate the effectiveness of the LSEEK filter and the upgraded observation

operator in assimilating a multivariate set of satellite SSH, foundation SST, T/S Argo profiles and daily skin SST measurements

at noon time. One challenge concerns the applicability of the new operator (designed for the Med-MFC z-level modelling

system) for sigma-level models, which inevitably limits its operability only in model domain areas where the first sigma level

depth is equal or shallower than the minimum depth of the temperature profile assumed by the operator. For the case of the

Aegean Sea model, this restriction excludes the usage of skin SST observations located in areas where the total depth is 470

m or more and is expected to limit the effectiveness of skin SST data assimilation into the model.

Weekly assimilation of satellite SSH, gridded foundation SST data and Argo T/S profiles into the modelling system for a

12 month period Jan 2014 – Dec 2014 (model control run), reaffirmed the ability of the LSEEK filter to fit the assimilated data

within the specified uncertainties. The additional assimilation of daily skin SST SEVIRI data at 12:00 UTC for the same period

using V2 of the Observation Operator (EXP2) showed the impact of this data set in correcting observed and unobserved

variables of the modelling system. In particular, the significant improvement obtained in the SSH analysis fields due to the

introduction of noon time skin SST observations is partially maintained in the model forecasts implying that the data

assimilation system is capable of correcting, in a dynamically consistent way, the initial errors responsible for later forecast

error growth.

It is also interesting to note that the quite frequent re-initialization of the model due to the assimilation of skin SST on a

daily basis data does contaminate the model solution due to the inevitable adjustment of the model dynamics to the analysis

fields. In the two specific examples shown and discussed the assimilation of skin SST is able to correct the shape, strength and

positioning of important dynamical features of the south Aegean and the central Levantine basin. On the other hand,

considering the behaviour of the analysis and forecast skin SST and the overall (2 m - 1000 m) temperature RMS error in

relation to the number of assimilated skin SST observations within the year, it becomes evident that the inherent limitation of

the new observation operator when applied to a sigma-level model has some effect on the skill of skin SST data assimilation



on the model dynamics. To this end, an additional year of data assimilation could be useful in order to understand if the rather poor performance of the data assimilation system observed between Jan-April 2014 is a result of the initial adjustment of
system to the newly introduced skin SST observations or it is exclusively attributed to the limited number of skin SST observations assimilated until April 2014.
To conclude, we have shown the feasibility of embedding low-rank statistics from a dynamical model, GOTM in our case, to formulate a dynamically consistent and statistically based observation operator capable to handle SST observations in a data
assimilation system aware of the diurnal SST variability. The methodology is quite general and can be applied in the future to a large variety of applications where complicated processes may be simplified into reduced-order observation operator, such
as for instance coupled ocean-sea-ice or ocean-biogeochemistry assimilation problem. Additionally, future work can be devoted into formulating an SST observation operator adjusted directly to sigma levels space in order to avoid exclusion of
SST observations in geographical areas where the operator profile is not compatible with temperature profile assumed by the model.
**Acknowledgments**

This work forms part of the SOSSTA project (http://sossta.cmcc.it/) which has been funded by the EU Copernicus Marine
Environment Monitoring Service (CMEMS) through the Service Evolution grants and it has been conducted using CMEMS information. S. Pimentel and W-H. Tse were additionally supported by the Natural Sciences and Engineering Research Council
(NSERC) of Canada. The GOTM simulations in this study used computing resources provided by WestGrid (www.westgrid.ca) and Compute Canada (www.computecanada.ca). Finally, Argo data were collected and made freely
available by the International Argo Program and the national programs that contribute to it. (http://www.argo.ucsd.edu, http://argo.jcommops.org). The Argo Program is part of the Global Ocean Observing System.

The CCA software was developed in python and is available upon request to the authors.

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
