# Peer review of "Assimilation of SST data in the POSEIDON system using the SOSSTA statistical-dynamical observation operator"

_Ocean Science, 2018_

## Referee Comment (RC1) · Anonymous Referee #1 · 7 Feb 2019

**1   general comments**

The manuscript "Assimilation of SST data in the POSEIDON system using the SOSSTA statistical-dynamical observation operator" posits a new observation operator (the SOSSTA operator) that accounts for diurnal warming of the water column in the near surface. The suggested method is statistical and uses pre-estimated functional forms, which depend on the wind speed and insolation, to extrapolate the model temperature to the surface. The extrapolated observations are then compared to satellite observations, with the observation minus model differences used in a data assimilation scheme. Results from a test of the system are shown using a model of the Aegean

sea.

I found the scientific ideas in this paper to be reasonable and worth pursuing. I also, despite the clear need for additional proof reading (eg every figure reference in the text was wrong), found the paper to written to a reasonable standard. Unfortunately, I believe the methodology used when actually applying their ideas to be seriously flawed. In particular, the decision to throw away the data over most of the domain is questionable. Additionally, the authors have also failed to show that their method has any benefit over just using a simpler observation operator—an absolutely vital requirement for a paper of this type.

To fix the problems I believe the authors will need to rerun all of their experiments and include at least one more. Obviously this will require a rewrite of the results section as well. This will be a very substantial amount of work and probably not feasible as part of this submission of the paper. I am thus recommending the paper be rejected. However, as I believe the underlying ideas to be sound, a resubmission of the paper at a later date should be acceptable if actions are taken to remedy the identified issues.

**2  specific comments**

- The single biggest flaw with the manuscript is the lack of a control experiment where the SEVIRI data are assimilated by just comparing the SST values directly with the top model level (as is done with the AVHRR data). The authors scheme is supposed to be an improvement on this simple method, so not including such a test is a massive omission.

- The justification for discarding SEVIRI data where the top model level is deeper than 1.55 m is poor. It is clear from their figure 7 that this choice has resulted in most of the data in the domain simply being thrown away, and it is difficult to see this as anything other than a major flaw with the method. It is a reasonable

assumption that the night time diurnal signal is small so, at the very least, night time SEVIRI data should have been used everywhere.

- I also think that excluding AVHRR data when the model level is deeper than 6 m is a poor decision, and throws away significant numbers of perfectly good observations. Firstly, the choice of 6 m seems arbitrary and is not justified (GHRSST tend to use 10 m as representative of the foundation temperature). Secondly, once you get below the warm layer, the temperature should be fairly constant down to the base of the mixed layer. I don't know the depth of the mixed layer in the study area, but unless it is shallower than 6 m the observations should be used.

- From reading the paper it is clear that the canonical operators depend on the atmospheric conditions. However, this wasn't really apparent until the results section. I think this fact needs to be given much greater prominence and discussion earlier in the paper. Also, how many sets of operators were produced, and what were the associated weather conditions?

- (L92-93) I found it surprising that the RMS errors against SEVIRI directly were smaller than the diurnal magnitude errors (which is difference between two noisy numbers, with additive errors). This is probably correct, but needs further comment.

- (Sec 2.1) You talk about the RMS error, but it would be good to see how the error splits into bias and random parts.

- (Sec 2.1) I don't see the point in the first training set. Using the average temperature rather than the morning temperature will bias everything warm. To my mind this is not scientifically justifiable, so should not be included.

- (Sec 2.2) I think this section should be re-written to make things clearer. In particular:

- You seem to have written the equations assuming a single observation (although you don't state this). In reality this will rarely be the case. It would be better to consider the multiple observation case and state the size of the vectors and matrices.

- In general your $\mathbf{a}$ and $\mathbf{b}$ will be matrices, not vectors, and so should be in capitals. Likewise, your $\mathbf{X}$, and $\mathbf{Y}$ are vectors and should be in lower case.

- It doesn't really matter, but it is more common to write the observation operator as a left hand multiplication, rather than a right hand multiplication.

- $\mathbf{U}$ and $\mathbf{V}$ are not sufficiently described. I believe that they are model and observation measures of the same canonical variable, but more time should have been spent describing them.

- Does $\mathrm{trace}(\mathbf{D}) = 1$? I think it does, but it's not clear and should be stated.

- on Line 112 you state that the intercept has to be zero. It is not obvious why this is the case, why can't the intercept be non-zero?

- When you write 'mean' be clear if you mean the time mean or space mean.

- (Fig 1) Show all of the SEVIRI data, and hash out the area deeper than your cut off.

- (Sec 4) More details need to be given about the quality control procedures. In particular I am not clear if different observations were rejected in each of the experiments. The aim of the paper is not to test the QC, so the same observations should have been used in every experiment.

- (Sec 4.1.2) More discussion is needed here. The aim of the proposed method is to improve results when there is a large diurnal cycle in SST. However, this section seems to show that the proposed method produces the best improvement in winter when the diurnal cycle is small. The authors need to put more effort into

explaining this contradiction (which, I believe, arises because of the lack of a proper control experiment).

- (Sec 4.2.1) The SEVIRI data cannot be used as independent validation of your system because it is assimilated in your experiments using the SOSSTA system. This is true even if SEVIRI is not used in your control. What your results show is that assimilating SEVIRI data brings the model closer to SEVIRI data, which is not a surprising result. Better questions to ask are: Does it degrade/improve the results to other data sources? What are the impacts on the forecasts. Are there genuinely independent data sources that can be used? Is it worth withholding data to use for validation?

- (Line 334) Do you really only assimilate SEVIRI data if no other observations are available? This seems wrong.

- (Line 345) I can see no justification for only assimilating SEVIRI data if you have more than 200 observations. After QC you should assimilate everything that is left, even if that is only a few observations.

- Too much focus is given in the results section on the change in the analysis with respect to the assimilated variables. The analysis should be closer to the assimilated observations, if it's not then something is very wrong indeed. What is more interesting is the effect on the forecast (including the background) and comparisons to independent observations.

**3   technical corrections**

- The figure numbering has gone badly wrong. It's fine in the captions, but wrong everywhere in the main text.

- (L18) In the abstract I don't understand what you mean by misfits. Do you mean the innovations?

- (L20) state what specific skill scores you use. Just saying 'skill scores' is not informative.

- (L28) "weather and climate" not "weather or climate".

- (L33) Rewrite as "...through to the present with the launch of several satellite missions, both polar orbiting and..."

- (L37) "10 $\mu$m depth" not "10 $\mu$m of depth".

- (L29) Reynolds is not an analysis of foundation SST. It is an estimate of the daily mean SST, which is not the same thing. OSTIA, however, is a foundation estimate.

- (L43) "in model physics" not "in the model physics".

- (L46) I think you need more than one example if you're going to say 'majority'.

- (L50) The While et al 2015 reference is to a conference proceedings not a paper. The work is covered in the 2014 and 2016 while et al references, so the 2015 reference should be removed.

- (L54) Spurious bracket before coupled.

- (L55) SST data are observations, they are not an observing network—this should be reworded.

- (L62) "with their model equivalent" not "with model equivalent".

- (L66) I don't think there should be brackets around "skin and subskin". Incorporate this directly into the sentence.

- (L74) "of the SOSSTA" not "of SOSSTA".

- (L93) At this point in the paper you have not defined what you mean by diurnal magnitude (I know it's defined later in the paper). It is therefore unreasonable to expect the reader to know what it is.

- (Sec 2.1) Rewrite this section. It is not clear if your second training set uses average temperature or morning temperature for initialisation.

- (L109) I think you should say "combination of skin SST and subsurface SST", not just "combination of SST"

- (L230) "was also" not "were also".

- (Everywhere) Do not use the term "offline" to mean "without data assimilation".

- (L240) What does V3 differ from?

- (Sec 4) It will be easier for the reader if you summarise the differences between V1,V2,V3, and V4 in a table.

- (L312) Comma after 'performance'.

- (Fig 3 caption) Write the criterion in full in the caption.

- (L362) I don't think "Error! Reference source not found" should appear.

- (L323) Don't count time using just days containing SEVIRI data and ignoring gaps, it is likely to confuse.

- (L446) "as is shown below" not "also as it will be shown below"

- (L471) "SSH which, with respect to the other model state variables, varies on shorter" not "SSH which with respect to other model state variables varies on shorter". Note the commas in my version; they mark out the sub clause and are important.

- (Figs 9 and 11) Please include difference plots as well as images of the absolute values.

- (L500) "...SSH, other model variables..." not "...SSH, also other model variables"

---

## Referee Comment (RC2) · Anonymous Referee #2 · 3 Apr 2019

General comments

This article addresses a very specific problem for model - observation comparison of satellite SST. This question is of primary importance when assimilating skin/sub skin satellite SST to produce ocean analysis and forecasts, but also for model evaluation.

This article presents and evaluates a quite complex but promising observation operator allowing the computation of the equivalent of a satellite skin and/or sub skin SST observation from an OGCM simulation. The estimation of the skin/subskin SST is based on statistical analysis using a 1D turbulence model. Today, most of OGCM do not have a vertical resolution that allows representing skin / sub skin temperature, the first vertical level thickness being in the order of a meter or 50 cm. The differences between the foundation, skin, sub skin and model first level temperature can be larger than 1°C under certain atmospheric conditions and this prevents an efficient assimilation of the satellite SST observation if just compared with the 1st vertical model level temperature.

An offline evaluation is first done comparing SEVIRI SST innovations computed using the 1st model temperature and the SOSSTA observation operator in the Aegean region. A better fit to the observations is obtained with SOSSTA, especially in summer when the diurnal cycle is larger, showing the benefit of the use of the SOSSTA operator versus the 1st OGCM level temperature.

Then experiments using the SOSSTA operator to assimilate SEVIRI skin SST in the Aegean region are presented.

The article is well written. The offline validation approach and the implementation in a data assimilation framework is well exposed. The analysis of the results is clear and goes from the validation of the assimilation process itself to the physical analysis of the region of interest, ie the Aegean Sea.

The originality of the SOSSTA observation operator compared to other approaches used today should be exposed.

The weakness of this article lies in the evaluation of the SOSSTA observation operator in the data assimilation framework. Comparison is made with simulations assimilating or not SEVIRI SST daily, considering weekly assimilation window. The comparison shows a very significant improvement of the analysis and forecasts, not only restricted to the additionally "observed" variable (the SEVIRI SST) but also the SSH and the full integrated temperature profile. As mentioned in the text, it is not possible to identify the improvements brought by the daily assimilation of SST, the use of SEVIRI SST with high spatial resolution or the use of an improved SST observation operator.

An additional experiment, with SEVIRI daily skin SST assimilated using for example

the first level model temperature as for the offline validation, is required to evaluate properly the improvement from the SOSSTA observation operator itself.

The use of sigma "model" also strongly limit the area where skin SST is assimilated due to too thick 1st layer but this point is well explained and taking into account when analyzing the results.

I would recommend a major revision before publication to overcome the lack of a proper evaluation of the SOSSTA observation operator itself in a data assimilation context.

Specific comments

- l. 63 : define explicitly the term "innovation" as it is appears for the first time in the paper here.

2.1 GOTM simulations

- 3 settings for learning: explain the physical differences that are expected to bring significantly different responses when using the operator.

2.2 CCA

- L. 124: Can you briefly justify why you choose to have a dependency of the transformation M to the wind and insolation.

3.3 Observational datasets

- L. 194: Can you give more precise description on the area where the foundation SST is not assimilated due to the thickness of the first layer?

- L. 204: the SEVIRI skin SST observations are assimilated keeping the original spatial resolution, ie 4.5km?

4.1 Offline evaluation

- Recall that the evaluation is done with the 3h frequency SEVIRI data as mentioned before l.201.

- L.296 and figure 2: The seasonal variability of the improvements due to the use of the SOSSTA seems to be counterintuitive: in winter the 1st level and skin SST may not differ as strongly as in summer with large heat fluxes and low wind conditions. Could you discuss this point?

- L.306: do you have any physical explanation for the better performance of the v2 obs operator version compared to the 3 others?

4.2 Online evaluation

Does the assessment still done with the 3h SEVERI SST?

- L344: ii) What do you mean by "on the days where no other observations are assimilated"? What is the consequence? Is the innovations for the different datasets are computed at the time of the observations (FGAT)?

- L.345: iii) …higher than 200. Is it a value computed over the week for a model grid cell / analysis point?

- L.349: what is the most stringent criteria in eliminating the number of assimilated observations?

- L.362: missing reference

- p.14: Table 3: Even if the SEVIRI data are not assimilated in the CTRL experiment, you still can diagnose the misfit to the analysis that can have a value lower than the innovation due to the multi-obs / multivariate properties of the DA system.

- P.19: Figure 7: the color bar is too large to highlight the improvements in most of the domain.

Technical corrections

- The reference to the figure numbers is wrong all along the text.

- The axis labels on the figures are most of the time too small. (fig. 16,…)

---

## Editor Comment (EC1) · Angelique Melet (Editor) · 14 May 2019

The manuscript is rejected as it requires substantial new analyses and rewriting to address the issues raised by the reviewers, not feasible as part of the present submission. Yet, as both reviewers recognized the scientific ideas of the paper as reasonable and worth pursuing, the authors are encouraged to submit a new version of their manuscript as part of a new submission.